# Axiomatic Software Design: What GenAI Cannot Design

## Abstract

The Abstract Reasoning Corpus (ARC) challenge highlights the persistent gap between current Artificial Intelligence (AI) systems and human-level reasoning, as even the most advanced Large Language Models (LLMs) struggle to match human performance—particularly in abductive reasoning, despite their growing strength in inductive and deductive tasks. This limitation is especially relevant in domains such as software design, where effective system creation requires abstract thinking, abductive hypothesis formation, and deductive synthesis, underscoring the broader challenge of achieving truly human-like reasoning in AI.

This study demonstrates how a systematic design framework, namely axiomatic design, can help mitigate weaknesses in AI-augmented software engineering.

**ACM Reference Format:**
Anonymous Author(s). 2026. Axiomatic Software Design: What GenAI Cannot Design. In *Proceedings of 3rd ACM International Conference on AI-powered Software (AIware 2026)*. ACM, New York, NY, USA, 10 pages. https://doi.org/XXXXXXX.XXXXXXX

## 1 Introduction

The 2021 report by the Carnegie Mellon University Software Engineering Institute highlights six major research areas that are expected to shape software engineering over the next 10 to 15 years. Among these research areas, Artificial Intelligence (AI)-augmented software development has attracted the most attention in both academia and industry[1, 17, 45].

The aim of *AI-augmented software development* is to increase developer productivity and reduce cognitive load by keeping humans focused on higher-level conceptual work while leveraging automation to reduce errors in tasks where machines can effectively assist humans [23, 38].

However, AI-augmented software development has primarily targeted *technical debt*, *refactoring*, *requirements traceability*, and automated *code generation* [23, 28, 38, 57, 63], yet it largely overlooks fundamental problems such as poorly specified requirements that result in suboptimal design.

Current trends in LLMs4SE (Large Language Models for Software Engineering) still treat requirements as a separate, sequential phase [35]. In a 2024 survey [20], only 3.9% of studies applied LLMs to requirements engineering, and just 0.92% targeted software design, whereas 56.65% focused on software development (i.e., code generation) and 22.71% on software maintenance (i.e., refactoring), with the remainder spread across other areas.

The disparity we observe between requirements engineering, software design, and software development (i.e., code generation) exists because code generation is a largely syntactic, translation-based task that is well-suited to the strengths of LLMs (i.e., code generation is a well-bounded problem), whereas requirements engineering and software design are profoundly semantic, creative, and analytic activities that require abstract thinking [60].

Abstract thinking is a cognitive process that focuses on the superordinate and general characteristics of an event. This form of thinking can enhance creativity because it enables individuals to generate novel and even unprecedented solutions [44, pp. 267].

The Abstract Reasoning Corpus (ARC) challenge represents a significant benchmark for developing Artificial Intelligence (AI) systems capable of human-like reasoning. Even the most advanced Large Language Models (LLMs) fall short of human performance [27]. While prior work has sought to close this gap [39], at the time of writing, it still seems unlikely that state-of-the-art LLMs can reliably elicit correct requirements and decompose them into the kind of hierarchical structure required by ISO/IEC/IEEE 29148:2018 and ISO/IEC/IEEE-24765:2017 [21, 22].

Therefore, the low adoption rates that we observe in the adaptation of LLM in software requirements engineering reflect not a lack of opportunity but a rational assessment of the current technology's fit for a deeply complex problem. For instance, the LLM's task is to translate a relatively clear prompt (e.g., "write a Python function to sort a list using quicksort") into a syntactically and logically correct output in Python. Requirements Engineering, conversely, deals with ambiguity, conflicting stakeholder needs (Arrow's impossibility theorem [3]), and emergent knowledge. It's about discovery, negotiation, and refinement, not just translation. At the same time, as Foundation Models (FMs) increasingly power tools that can produce functionally correct code, systematic design methods that tackle challenges such as requirements engineering—and that can guide AI design assistants—are likely to receive renewed emphasis.

It seems that AD can potentially be a great help for co-engineering software systems by assisting software designers in better guiding the systematic decomposition of functional requirements into coherent design parameters in collaborative settings where human engineers work alongside LLM-based AI design assistants (e.g., Copilot, Cursor), thereby maintaining independence between concerns, preserving traceability across abstraction levels, and structuring the interaction between human judgment and machine-generated proposals.

However, the end-to-end application of Axiomatic Design (AD) in software design has not been studied comprehensively since its early introduction [26, 53]; for instance, the majority of works overlook the process domain of AD [47] and, consequently, process variables [9, 11, 15]; some that do address it propose interpretations that override its originally intended role [48]. Others argue that AD is inappropriate for software products [16, 43], although many

of these criticisms are addressed in [4]. So far, even those who advocate applying AD to software have stopped short of generating compilable, executable code, and only a few studies have reached even an initial high-level software design.

This study highlights AD's potential to bring greater rigor to software design and, in doing so, help mitigate weaknesses in AI-augmented software engineering. Specifically, unlike earlier efforts [9, 15, 19, 32, 53], this paper demonstrates an end-to-end application of AD to software systems, spanning both high-level design (element interaction) and low-level design (element internals). This is enabled by a software-centric, *hierarchical* interpretation of "design parameters (DPs)" and "process variables (PVs)" that organizes their original definitions across abstraction levels rather than replacing them. Second, we make the *process domain* an explicit part of the software design workflow while remaining faithful to its original definition. Third, we introduce syntaxes for FRs and DPs that help designers—and LLM-based AI design assistants—avoid conflating the "what" domain with the "how" domain.

To keep the focus on the core ideas and avoid the overhead of a complex running example, we use a simple calculator program from [42] that performs the four basic arithmetic operations (+, −, ×, ÷). This keeps both the example and its implementation easy to follow. The example is used solely to demonstrate how AD can be applied in practice, from *needs* to *code*, and is not intended as a comparison with the work of Proulx et al. [42].

In addition, we have moved the theoretical introduction to AD to an appendix. This allows us to demonstrate how, once AD thinking is grasped, it can guide the designer in practice without requiring the designer to *engage directly* with the underlying mathematics all the time.

The remainder of this paper is organized as follows. Section 2 briefly reviews axiomatic design. Section 3 introduces the FR and DP syntaxes. Section 4 applies axiomatic design to a running example, and Section 5 presents the corresponding code. Section 6 discusses our new view on DPs and PVs. Section ?? outlines what axiomatic design can offer, and Section 8 concludes the paper.

## 2 Briefing on Axiomatic Design

Unlike in design theory [8, 51, 61], people in software design generally assume that design begins only after the requirements have been established. This view, originating in "Structured Design" [62, pp. 16], has persisted to the present day—explicitly in some works [7, pp. 25][49, pp. 197][41, pp. 156][10, pp. 4] and implicitly in others [6, pp. 1-3].

In contrast, *axiomatic design*, originally introduced for manufacturing systems in 1978 [52] and later adopted in many other engineering fields [33, 36], relies on design theory by treating the definition of requirements as the main part of the design process. As a result, the design activity begins before the requirements are established.

AD operates across four domains (see Fig. 1), usually viewed from left to right as follows: the *Customer Domain* (identifying needs: CAs - Customer Attributes - i.e., Stakeholder Requirements), the *Functional Domain* (defining independent functional requirements, FRs), the *Physical Domain* (determining design parameters, DPs, that satisfy FRs), and the *Process Domain* (establishing process variables, PVs, to generate DPs).

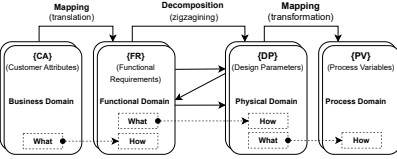

**Figure 1: Four design domain starting from left (the "What" domain) to right (the "How" domain) [33].**

In AD, designers systematically interplay among design domains, considering the domain on the left as "what we want to achieve" and the domain on the right as "how we want to achieve the what". This process is called *zigzagging*. However, to begin zigzagging, we must first define the first-level FRs (i.e., problem formulation).

Therefore, the problem formulation in terms of first-level FRs remains the most critical stage in the design process because a poor problem definition leads to unacceptable or unnecessarily complex (i.e., under-engineered or over-engineered) solutions [50, pp. 30-32]. This view is confirmed by others[29, 30]. For example, for a road vehicle, problem formulation could result in the following first-level FRs: *shall go forward, shall go backward, shall turn,* and *shall stop* [33, pp. 21].

### 2.1 Design Equation

Functional requirements (FRs) that define specific design goals are represented as the FR vector in the functional domain. Similarly, the set of design parameters (DPs) in the physical domain, chosen to satisfy the FRs, constitutes the DP vector. The relationship between these two vectors can be expressed as [33, pp. 18]:

$$\{FR\} = [A]\{DP\} \tag{1}$$

where *[A]* is a matrix (referred to as the design matrix) that relates FR to DP.

$$[A] = \begin{bmatrix} A_{11} & A_{12} & A_{13} \\ A_{21} & A_{22} & A_{23} \\ A_{31} & A_{32} & A_{33} \end{bmatrix}$$

When the Eq. 1 is written in differential form as

$$\{dFR\} = [A]\{dDP\} \tag{2}$$

the elements of the design matrix are given by

$$A_{ij} = \frac{\partial FR_i}{\partial DP_j} \tag{3}$$

With three FRs and three DPs, Eq 1 may be expressed in terms of its elements as

$$FR_i = \sum_{j=1}^{3} A_{ij} \cdot DP_j \tag{4}$$

or

$$\begin{aligned} FR_1 &= A_{11}DP_1 + A_{12}DP_2 + A_{13}DP_3, \\ FR_2 &= A_{21}DP_1 + A_{22}DP_2 + A_{23}DP_3, \\ FR_3 &= A_{31}DP_1 + A_{32}DP_2 + A_{33}DP_3. \end{aligned} \tag{5}$$

and, in its general form, it can be formulated as:

$$FR_i = \sum_{j=1}^{n} A_{ij} \cdot DP_j \qquad (6)$$

where $n$ = the number of DPs. For designing a process (i.e., mapping from the DP in the physical domain to the PV in the process domain), the equation will be

$$\{DP\} = [B]\{PV\} \qquad (7)$$

where $[B]$ is the design matrix that defines the characteristics of the *process design* , similar to $[A]$. The elements of the design matrices $[A]$ and $[B]$ can be either constants or functions. If the elements of the matrix are constants, the design is linear, while if the elements are functions of DPs, the design may be non-linear [33, pp. 20].

## 2.2 Independence Axiom

The Independence Axiom defines *coupling* in terms of the "what" vs. the "how". for example, each FR should be satisfied without affecting others; i.e., a vector of DPs should satisfy only one FR and no more. This is achieved when the design matrix relating FRs and DPs is either diagonal (uncoupled design - one DP per FR, see (a) in Fig. 2) or triangular (decoupled design, a set of DPs per FR, see (b) in Fig. 2), whereas a fully populated matrix indicates a coupled and undesirable design.

$$\begin{bmatrix} A_{11} & 0 & 0 \\ 0 & A_{22} & 0 \\ 0 & 0 & A_{33} \end{bmatrix} \quad \begin{bmatrix} A_{11} & 0 & 0 \\ A_{21} & A_{22} & 0 \\ A_{31} & A_{32} & A_{33} \end{bmatrix} \quad \begin{bmatrix} A_{11} & A_{12} & A_{13} \\ A_{21} & A_{22} & A_{23} \\ A_{31} & A_{32} & A_{33} \end{bmatrix}$$
$$(a) \qquad\qquad (b) \qquad\qquad (c)$$

**Figure 2: Each row corresponds to an FR and each column to a DP. Matrices (a), (b), and (c) illustrate uncoupled, decoupled, and coupled designs, respectively.**

## 2.3 Zigzagging

Zigzagging is an iterative hierarchical decomposition process in the sense that the designer can return to the domain on the left based on the ideas generated in the domain on the right [33]. This helps increase knowledge about the system, make more informed design decisions (i.e., avoid speculation), postpone decisions to the last responsible moment (lean methodology [31]), and evaluate each design decision layer by layer, making it a great aid in avoiding both over-engineering and under-engineering.

Zigzagging helps to decompose the system layer by layer. This iterative and incremental decomposition allows the design to be evaluated at each layer. At each decomposition level, the design equation could ensure consistency with the high-level intention and evaluate adherence to the Independence Axiom [33, pp.29]. For example, if the Independence Axiom is not satisfied, the designer must either stop and conduct research (e.g., use innovation) to develop a new set of FRs and/or DPs that will satisfy the Independence Axiom [33, pp. 51].

Zigzagging separates "what" from "how" at every level, aligns domains and levels of abstraction, and enables traceability and

change impact reasoning. It illustrates how the chosen DP in the physical domain directly shapes the resulting sub-FRs (see Fig. 3).

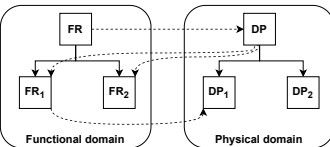

**Figure 3: Zigzagging to decompose FRs and DPs [33, pp. 30].**

This decomposition process ends once the designer captures enough information to create the system [33, pp.29-30].

## 2.4 Theorems and Corollaries

In axiomatic design, there exists a list of corollaries (e.g., Corollary 2 – minimize the number of FRs) and theorems (e.g., Theorem S3 – Importance of High-Level Decisions) to support the designer. However, in this study, we do not discuss any theorems or corollaries in detail. The most recent list of existing corollaries and theorems in AD can be studied in Suh's work on complexity theory [34, pp. 44–51].

## 3 Syntax for FR and DP Formulation

For the first time, we propose the following syntaxes for formulating functional requirements (FRs) and design parameters (DPs), grounded in ISO/IEC/IEEE 29148 [22] and the distinction between input and system constraints. In ISO/IEC/IEEE 29148, two main syntaxes are defined for specifying requirements:

*[Subject] SHALL [Action] [Constraint of Action]*
*[Condition] [Subject] SHALL [Action] [Object] [Constraint of Action]*

## 3.1 Syntax for FR

We argue that the first syntax proposed by ISO/IEC/IEEE 29148 is a good candidate for first-level FRs. At this level, FRs must be strictly solution-neutral, minimal, independent, and abstract—capturing only the fundamental "what" and being bounded solely by input constraints (ICs), which are typically expressed as design ranges. The first syntax does exactly this: it states only the required function, with the "Constraint of Action" mapping cleanly to AD's ICs, and it avoids premature decomposition, whereas the more detailed second syntax risks incorporating conditions, flows, states, and logic (i.e., the "how") into the FR too early.

The second syntax is more appropriate after zigzagging when FRs become more operational, when preconditions and interactions matter, when sequencing or state-based constraints emerge, and when the first design matrix structure is already established.

At these lower levels, it becomes necessary to state when the function should be active (condition), what it acts on (object), and what tolerances are allowed (constraint). These lower level tolerances often stem from DPs chosen at higher levels; we refer to them as system constraints.

## 3.2 Syntax for DP

In the case of the DP formulation, we suggest using the following syntax. For the high-level DPs, we suggest using the first one, and for the next level DPs, the second one:

[Object] [Operation on Inputs] to [Produce Outputs]

[Object] [Perform Atomic Operation] to [Produce Atomic Output]

Depending on the decomposition layer, the *Object* could be a system, sub-system, service, component, module, algorithm, method, function, or a single input parameter.

## 4 Example: Design of a Calculator

To clarify the ideas, we adopt an example from [42]: a simple calculator without a user interface. The calculator is intended to perform the four basic arithmetic operations $(+, -, \times, \div)$ on two *natural numbers*[1].

## 4.1 Step 1: Problem Formulation

As discussed earlier, AD requires that CAs be translated into a *minimal*, *independent*, and solution-neutral set of first-level FRs [33, 50]. This minimizes cost and complexity, supports incremental and anti-coupled development [4, 34], and avoids bias while preserving innovation. Each FR is then refined by zigzagging between the *functional* ("what") and *physical* ("how") domains: one FR is taken at a time, a set of DPs is selected to satisfy it, and the process returns to the functional domain to decompose it into sub-FRs [33, 50].

Table 1 lists the resulting first-level FRs.

### Table 1: First-Level FRs

| FR | Description |
| --- | --- |
| $FR_1$ | The Calculator SHALL accept two operands from natural numbers and one operator from $\{+, -, \times, \div\}$. |
| $FR_2$ | The Calculator SHALL produce the result of applying the received operator to the two operands. |

## 4.2 Step 2: Zigging to Chose DPs

After establishing the highest-level FRs, the next step is the conceptualization process, which involves "zigging" from the functional domain to the physical domain to determine how each FR will be realized. Table 2 represents potential DPs that could satisfy $FR_1$ and $FR_2$.

### Table 2: First-Level DPs

| DP | Description |
| --- | --- |
| $DP_1$ | Intake module that accepts two *32bit integers* pointers as operands and a pointer of length *one byte* as operator to read operands and operator and write them into memory addresses. |
| $DP_2$ | Computing module that applies the received operator to the two *32bit integers* values to produce one *32bit integers* result. |

Note how moving into the physical domain already captures "how" information: numbers are restricted to *32bit integers*, and only predefined operators with *one byte* length is permitted. All of

---

[1]Division with remainder and large multiplication ($\text{int32} \times \text{int32}$) results are taken out of scope.

these "how" aspects narrow the original "what" in the next level of decomposition by introducing additional constraints that will be accommodated by the next level of FRs.

At this stage, the first layer of design for $FR_1$ and $DP_1$ is complete, and we can evaluate the design by creating the design matrix:

$$\begin{bmatrix} FR_1 \\ FR_2 \end{bmatrix} = \begin{bmatrix} X & 0 \\ X & X \end{bmatrix} \begin{Bmatrix} DP_1 \\ DP_2 \end{Bmatrix} \tag{8}$$

In terms of the product design matrix, we see a lower-triangular form that is decoupled when instantiated in the sequence $DP_1 \rightarrow DP_2$.

## 4.3 Step 3: Zagging to Decompose $FR_1$

After selecting the first-level DPs by "zigging" from the functional domain into the physical domain, the next step is to "zag" (returning back) to the functional domain and, based on the chosen DPs, further decompose FRs into sub-FRs. Remember that the objective is to define sub-FRs and sub-DPs in a manner that ensures the fulfillment of the objectives specified by their corresponding higher-level FR and DP. A possible next level of FRs may be defined as we see in the Table 3.

### Table 3: Decomposition of $FR_1$

| FR | Description |
| --- | --- |
| $FR_{1.1}$ | The Calculator SHALL detect the start and end of a request. |
| $FR_{1.2}$ | The Calculator SHALL receive the first *32bit* integer operand. |
| $FR_{1.3}$ | The Calculator SHALL receive the second *32bit* integer operand. |
| $FR_{1.4}$ | The Calculator SHALL receive one operator from the list of possible operators $\{+, -, \times, \div\}$; otherwise, it SHALL output an error. |
| $FR_{1.5}$ | The Calculator SHALL assemble the two *32bit* integers and the operator into one internal request representation. |
| $FR_{1.6}$ | The Calculator SHALL make the assembled request available to the Arithmetic Core. |

At first glance, sub-FRs presented in Table 3 appear to be a reasonable set of new FRs. However, a closer look shows that $FR_{1.1}$ (request boundary), $FR_{1.5}$ (request assembly), and $FR_{1.6}$ (handoff to computation) are all speculations about the "how", while we have not yet made decisions regarding their realization. Therefore, they should be removed from the set of sub-FRs in order to focus only on the "what". In addition, the formulation of $FR_{1.4}$ is not atomic, and this could result in either having the number of DPs greater than the number of FRs or necessitating one more step of decomposition on the $FR_{1.4}$. To address these issues, we will define a new set of sub-FRs following (see Table 4).

### Table 4: Re-Decomposition of $FR_1$

| FR | Description |
| --- | --- |
| $FR_{1.1}$ | The Calculator SHALL receive the first *32bit* integer operand. |
| $FR_{1.2}$ | The Calculator SHALL receive the second *32bit* integer operand. |
| $FR_{1.3}$ | The Calculator SHALL receive a *one byte* characther operator from the list of vlid operators $\{+, -, \times, \div\}$. |
| $FR_{1.4}$ | The Calculator SHALL validate the received operator. |

## 4.4 Step 4: Zigging to Chose sub-DPs

Based on the sub-FRs defined in Table 4, the next level of sub-DPs that could satisfy those sub-FRs may be chosen, as presented in Table 5.

**Table 5: Decomposition of DP$_1$**

| DP | Description |
|---|---|
| DP$_{1.1}$ | The intaker accepts a *32bit* integer pointer to read the first operand value from the input driver and write it into a 32bit address. |
| DP$_{1.2}$ | The intaker accepts a *32bit* integer pointer to read the second operand value from the input driver and write it into a 32bit address. |
| DP$_{1.3}$ | The intaker accepts a *char* pointer to read operator token from the input driver and write it into one byte of char address. |
| DP$_{1.4}$ | The intaker validates the operator and writes "\0" in the operator address and prints "Error: Invalid operator!" if the operator dose not belong to $\{+, -, \times, \div\}$. |

The second layer of design for FR$_1$ and DP$_1$ is complete, and we can evaluate the design by creating the design matrix:

$$\begin{bmatrix} FR_{1.1} \\ FR_{1.2} \\ FR_{1.3} \\ FR_{1.4} \end{bmatrix} = \begin{bmatrix} X & 0 & 0 & 0 \\ 0 & X & 0 & 0 \\ 0 & 0 & X & 0 \\ 0 & 0 & X & X \end{bmatrix} \begin{Bmatrix} DP_{1.1} \\ DP_{1.2} \\ DP_{1.3} \\ DP_{1.4} \end{Bmatrix} \quad (9)$$

The resulting design matrix (Eq. 10) is decoupled because there is an order among FR$_{1.3}$ and FR$_{1.4}$, meaning that to satisfy FR$_{1.4}$, we first have to satisfy FR$_{1.3}$.

At this stage, we have reached the leaf level for FR$_1$ and DP$_1$. We now proceed with FR$_2$ and DP$_2$.

## 4.5 Step 4: Zagging to Decompose FR$_2$

Based on the chosen DP$_2$, possible sub-FRs for FR$_2$ may be formulated as follows (see Table 6):

**Table 6: Decomposition of FR$_2$**

| FR | Description |
|---|---|
| FR$_{2.1}$ | When the operator is +, the Calculator SHALL output the sum of the two *32bit* integers. |
| FR$_{2.2}$ | When the operator is −, the Calculator SHALL output the difference of the two *32bit* integers. |
| FR$_{2.3}$ | When the operator is ×, the Calculator SHALL output the product of the two *32bit* integers. |
| FR$_{2.4}$ | When the operator is ÷ if the second *32bit* integer is zero, the Calculator SHALL output an error. |
| FR$_{2.5}$ | When the operator is ÷, the Calculator SHALL output the *32bit* integer result of dividing the first *32bit* integer by the second *32bit* integer. |

## 4.6 Step 5: Zigging to Chose sub-DPs

Now we zag into the physical domain and choose the sub-DPs for DP$_2$ based on the sub-FRs defined in Table 6.

**Table 7: Decomposition of DP$_2$**

| DP | Description |
|---|---|
| DP$_{2.1}$ | A summation algorithm that accepts exactly two *32bit* integers to produce one *32bit* integer as the sum. |
| DP$_{2.2}$ | A subtraction algorithm that accepts exactly two *32bit* integers and subtracts the second *32bit* integer from the first *32bit* integer to produce one *32bit* integer difference. |
| DP$_{2.3}$ | A multiplication algorithm that accepts exactlly two *32bit* integers multiply to produce one *32bit* integer product. |
| DP$_{2.4}$ | A division algorithm that accepts exactly two *32bit* integers, check the second *32bit* integer and, if it is zero, return a "divide by zero" error. |
| DP$_{2.5}$ | A division algorithm that accepts exactly two *32bit* integers divide the first *32bit* integer by the second *32bit* integer to produce one *32bit* integer quotient. |

The second layer of design for FR$_2$ and DP$_2$ is complete, and we can evaluate the design by creating the design matrix:

$$\begin{bmatrix} FR_{2.1} \\ FR_{2.2} \\ FR_{2.3} \\ FR_{2.4} \\ FR_{2.5} \end{bmatrix} = \begin{bmatrix} X & 0 & 0 & 0 & 0 \\ 0 & X & 0 & 0 & 0 \\ 0 & 0 & X & 0 & 0 \\ 0 & 0 & 0 & X & 0 \\ 0 & 0 & 0 & X & X \end{bmatrix} \begin{Bmatrix} DP_{2.1} \\ DP_{2.2} \\ DP_{2.3} \\ DP_{2.4} \\ DP_{2.5} \end{Bmatrix} \quad (10)$$

## 4.7 Step 6: Constructing DP$_1$ and DP$_2$

Having reached the leaf levels for FR$_1$/DP$_1$ and FR$_2$/DP$_2$ (product design complete), we now start a mapping process between the physical domain and the process domain. We adopt the following process-domain choices:

**Table 8: First-Level Process Variables**

| PV | Description |
|---|---|
| PV$_1$ | A public function with return type void named Intaker accepts two integer pointers of type int32_t and one pointer of type char. |
| PV$_2$ | A public function with return type of int32_t named evalute accepts two int32_t and a parameter of type char. |

Now that we have clarified how DP$_1$ and DP$_2$ are supposed to be constructed, we must build the process matrix and evaluate it against the Independence Axiom:

$$\begin{bmatrix} DP_1 \\ DP_2 \end{bmatrix} = \begin{bmatrix} X & 0 \\ 0 & X \end{bmatrix} \begin{Bmatrix} PV_1 \\ PV_2 \end{Bmatrix} \quad (11)$$

The design is acceptable and represents an ideal (uncoupled) design, as the construction of each DP does not depend on any other DP.

## 4.8 Step 7: Constructing sub-DPs of DP$_1$

Once the first-level PVs have been chosen, the design proceeds by constructing the sub-DPs. The following sub-PVs are selected according to the sub-DPs of $DP_1$.

**Table 9: Decomposition of $PV_1$**

| PV | Description |
|---|---|
| PV$_{1.1}$ | A static local function of return type void named intake_first_operand that accepts a pointer of type int32_t named "op". |
| PV$_{1.2}$ | A static local function of return type void named intake_second_operand that accepts a pointer of type int32_t named "op". |
| PV$_{1.3}$ | A static local function of return type void named intake_operator that accepts a pointer of type char named "op". |
| PV$_{1.4}$ | A static local function of return type void named validate_operator that accepts a pointer of type char named "op". |

Now that we have clarified how sub-DPs of DP$_1$ are supposed to be constructed, we must build the process design matrix (i.e., $[B]$) and evaluate it against the Independence Axiom:

$$\begin{bmatrix} DP_{1.1} \\ DP_{1.2} \\ DP_{1.3} \\ DP_{1.4} \end{bmatrix} = \begin{bmatrix} X & 0 & 0 & 0 \\ 0 & X & 0 & 0 \\ 0 & 0 & X & 0 \\ 0 & 0 & X & X \end{bmatrix} \begin{Bmatrix} PV_{1.1} \\ PV_{1.2} \\ PV_{1.3} \\ PV_{1.4} \end{Bmatrix} \quad (12)$$

As we can see, the design matrices $[A]$ (Eq. 9) and $[B]$ (Eq. 12) are identical.

## 4.9 Step 7: Constructing sub-DPs of $DP_2$

Table 10 lists the process variables that generate the sub-DPs of $DP_2$ based on the $PV_2$. This step marks the end of our design. Implementation can begin afterward. However, note that the resulting design matrix $[B]$ at this time does not comply with the Independence Axiom, even though it generates the same sub-DPs dictated by $[A]$. This further supports our claim that the process domain is equally important and must not be overlooked.

### Table 10: Decomposition of $PV_2$

| PV | Description |
|---|---|
| $PV_{2.1}$ | A static local function with return type int32_t named add that accepts two int32_t input parameters. |
| $PV_{2.2}$ | A static local function with return type int32_t named sub that accepts two int32_t input parameters. |
| $PV_{2.3}$ | A static local function with return type int32_t named mul that accepts two int32_t input parameters. |
| $PV_{2.4}$ | A static local function with return type int32_t named div that accepts two int32_t input parameters. |

The design matrix $[B]$ is coupled because $PV_{2.4}$ generates two closely related but distinct design parameters ($DP_{2.4}$ and $DP_{2.5}$). We intentionally postpone discussing how to resolve this coupling so that we can first show how it appears in the implementation, as we believe this may motivate future work[2].

$$\begin{bmatrix} DP_{2.1} \\ DP_{2.2} \\ DP_{2.3} \\ DP_{2.4} \\ DP_{2.5} \end{bmatrix} = \begin{bmatrix} X & 0 & 0 & 0 \\ 0 & X & 0 & 0 \\ 0 & 0 & X & 0 \\ 0 & 0 & 0 & X \\ 0 & 0 & 0 & X \end{bmatrix} \begin{Bmatrix} PV_{2.1} \\ PV_{2.2} \\ PV_{2.3} \\ PV_{2.4} \end{Bmatrix} \tag{13}$$

## 5 Resulting Artifacts

Fig. 4 presents the high-level module diagram, which shows two independent modules, each responsible for one first-level FR.

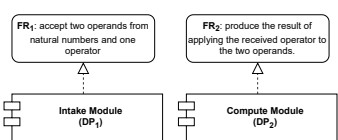

Figure 4: Conceptualization of $FR_1$ through $DP_1$ and $FR_2$ through $DP_2$.

The following is the complete implementation in $C$ code. In the $C$ programming language, the header file (*.h) represents process variables (PVs), while the source file (*.c) represents design parameters (DPs). The signatures of static and non-static functions, commands, and local variables correspond to PVs, whereas the function bodies, state machines, and control flows correspond to DPs. It is important to note that the entire code was developed by Copilot version 1.38 in VSCode by prompting the final FRs and DPs.

---

[2]In several respects, it represents a violation of cohesiveness [24, 62].

### Figure 5: Realization of ($PV_1$)

```c
1 #ifndef INTAKER_H
2 #define INTAKER_H
3 #include <stdint.h>
4
5 void intaker(uint32_t *op1, uint32_t *op2, char *op);
6 #endif // INTAKER_H
```

The intaker function, which spans lines 32 to 37 in Fig. 6, is, in fact, constructed by

$$DP_1 = \sum_{j=1}^{4} B_{j.1}PV_{1.j} + B_{j.2}PV_{1.j} + B_{j.3}PV_{1.j} + B_{j.4}PV_{1.j}. \tag{14}$$

### Figure 6: Intake module ($DP_1$): two int32 operands and one operator, with validation.

```c
1 #include <stdio.h>
2 #include <stdint.h>
3 #include <inttypes.h>
4 #include "intaker.h" // ensure this header declares void intaker
    (...)
5
6 /* DP_{1.1}: first operand intake */
7 static void intake_first_operand(uint32_t *op) {
8     scanf("%" SCNd32, op);
9 }
10
11 /* DP_{1.2}: second operand intake */
12 static void intake_second_operand(uint32_t *op) {
13     scanf("%" SCNd32, op);
14 }
15
16 /* DP_{1.3}: operator intake (note leading space to skip
        whitespace) */
17 static void intake_operator(char *op) {
18     scanf(" %c", op);
19 }
20
21 /* DP_{1.4}: operator validation */
22 static void validate_operator(char* op) {
23     int is_valid = (*op == '+' || *op == '-' || *op == '*' || *
        op == '/');
24     if (!is_valid)
25     {
26         fprintf(stderr, "Error: Invalid operator!\n");
27         *op = '\0'; // invalid operator
28     }
29 }
30
31 /* DP_{1}: intake all inputs and validate */
32 void intaker(uint32_t *op1, uint32_t *op2, char *op) {
33     intake_first_operand(op1);
34     intake_second_operand(op2);
35     intake_operator(op);
36     validate_operator(op);
37 }
```

The evaluate function, starting from lines 24 to 31 in Fig. 8, is, in fact, constructed by

$$DP_2 = \sum_{j=1}^{5} B_{j.1}PV_{1.1} + B_{j.2}PV_{1.2} + B_{j.3}PV_{1.3} + B_{j.4}PV_{1.4} + B_{j.5}PV_{1.4} \tag{15}$$

**Figure 7: Realization of ($PV_2$)**

```
1 #ifndef OPERATOR_H
2 #define OPERATOR_H
3
4 #include <stdint.h>
5
6 uint32_t evaluate(uint32_t op1, uint32_t op2, char op);
7
8 #endif // OPERATOR_H
```

In the above equation, $PV_{1.4}$ appears in the last two terms, which indicates coupling. We previously noted this issue but deferred addressing it, as we wanted to demonstrate how such coupling would appear in the implementation. In lines 13–21 of the code in Fig 8, this coupling can be observed directly in the implementation, with a code comment placed immediately above the corresponding function (div_q).

**Figure 8: Operator module ($DP_2$): two int32 operands and one operator to compute.**

```
1 #include <stdio.h>
2 #include <stdint.h>
3 #include <inttypes.h>
4 #include "operator.h"
5
6 static uint32_t add(uint32_t a, uint32_t b) {/*DP_{2.1}*/ return
    a + b; }
7 static uint32_t sub(uint32_t a, uint32_t b) {/*DP_{2.2}*/ return
    a - b; }
8 static uint32_t mul(uint32_t a, uint32_t b) {/*DP_{2.3}*/ return
    a * b; }
9
10 /*This code here is coupled because the PV_2.4 is generating two
        DPs (DP_2.4 and DP_2.5)
11  that are closely related to each other. This is violation of
        the Independence Axiom.
12 */
13 static uint32_t div_q(uint32_t a, uint32_t b) {
14   //DP_{2.4}: division by zero check
15   if (b == 0) {
16       fprintf(stderr, "Error: Division by zero\n");
17       return INT32_MIN; // Indicate error with a sentinel
      value
18   }
19   //DP_{2.5}: normal division
20   return a / b;
21 }
22
23 // DP_{2}: evaluate result based on operator
24 uint32_t evaluate(uint32_t op1, uint32_t op2, char op) {
25   switch (op) {
26       case '+': return add(op1, op2);
27       case '-': return sub(op1, op2);
28       case '*': return mul(op1, op2);
29       case '/': return div_q(op1, op2);
30       default: return 0; // Should not happen if operator is
      validated
31   }
32 }
```

To remove this coupling, our preferred approach, which is aligned with AD principles, is to return to $DP_{2.4}$ and reformulate it as follows:

$DP_{2.4}$: A **division validation** algorithm that accepts exactly two *32bit* integers, check the second *32bit* integer and, if it is zero, return a "divide by zero" error.

In fact, the design matrix in Eq. 10 was already coupled, but this went unnoticed because the DP titles differed ($DP_{2.4}$ and $DP_{2.5}$), even though both descriptions referred to the same design parameter ("A division algorithm"). This exposes a weakness of the proposed DP syntax; future work could refine it to reduce the likelihood of such issues, although it can never eliminate human error entirely. The most reliable safeguard is to carry the design through to completion, i.e., until the process domain.

## 6 Novel View on DP and PV

During our literature review [9, 12, 13, 15, 18, 25, 40, 46, 54–56], we identified that one of the main difficulties in the use of AD in the software domain is that, despite the step-by-step decomposition process (i.e., zigzagging), the functional requirements (FRs), design parameters (DPs), and process variables (PVs) are often interpreted in a flat, non-hierarchical manner, as originally introduced by Nam P. Suh [50]. This flat interpretation has made it difficult for practitioners to progress in decomposition while maintaining consistency across design domains, since DPs and PVs, in particular, can take on different roles depending on the level of abstraction.

Suh presented multiple interpretations of FRs, DPs, and PVs when designing a software product [50, pp. 245], which has led to confusion among researchers applying AD to software [2, 5, 14, 15, 37, 58, 59].

For example, Suh notes that a *module* may be treated either as a PV or as a DP [33, pp. 12], which is correct. However, he does not specify under what conditions it should be treated as a PV and when as a DP. We argue that a module should be regarded as a PV when it is used as an existing, out-of-the-box artifact to produce our DPs, and as a DP when the goal is to design that module itself. In other words, the distinction depends on usability: are we reusing an existing module (PV) to realize an intended DP, or are we developing the module as a new DP to satisfy a given FR?

In fact, most of Suh's original interpretations of DPs and PVs in software systems are valid, but they must be understood and applied according to the level of decomposition. For instance, in software design, treating DPs as inputs at the first level leads to premature design decisions that hinder further progress. Instead, DPs should be interpreted hierarchically: first as *modules*, then as *algorithms*, and only at the lowest level as *inputs*.

We, therefore, propose a hierarchical interpretation of FRs, DPs, and PVs in which their meanings shift with the level of abstraction as the design progresses down the decomposition tree. In our software-centric view, we do not discard the original meanings of DPs and PVs; rather, we organize their seemingly diverse interpretations into a coherent hierarchy. We now summarize this hierarchical interpretation of DPs and PVs in the context of a software project.

FRs retain their original definition:

- they represent an *output* at the lowest level of FRs (i.e., sub-sub-Functional Requirements),
- an atomic "what" at the mid-level,
- and a coarse-grained "what" at the highest level (top level Functional Requirements).

The distinction between fine-grained FRs (i.e., sub-FRs at the mid-level) and coarse-grained FRs (i.e., first-level FRs) lies in the amount of information encapsulated by each FR.

However, we interpret DPs differently than previous practices:

- the lowest level of DPs is interpreted as *inputs* intended to perform atomic operations.
- mid-level DPs are interpreted as an *algorithm*, *flowchart*, *state machine*, or a sequence of *basic functions* i.e., *commands* (e.g., ADD, SUB).
- DPs at the highest level of the design hierarchy are interpreted as a *system*, *subsystem*, *module*, or *component*.

In each case, the DP represents the "how" corresponding to the "what" defined by the related FR. Regarding our interpretation of PVs:

- The lowest level PVs represent local variables, function signatures, and/or interfaces.
- PVs at the higher levels of the decomposition hierarchy represent programming languages, sets of instructions (i.e., commands), libraries, compilers, and run-time environments.

These local variables, function signatures, and interfaces (e.g., APIs, data contracts), together with sets of instructions, must then be synthesized to generate a design solution (i.e., a design parameter) capable of producing the expected output to satisfy the intended FR. In other words, when all PVs (*set of instructions*, *interrupts*, *variables*) are combined, the intended design parameter emerges in the form of *custom functions*[3], *module*, or *class*.

Note that we distinguish between a set of instructions and a sequence of instructions. A set of instructions alone is meaningless and generates no value. However, when these instructions are arranged in a specific order to realize a flowchart or an algorithm (i.e., a design parameter), they form a sequence of instructions that generate values, i.e., expected behavior; this is regarded as DP.

This thinking aligns very well with the expectations we have for the zigzagging (step-by-step design) process, i.e., whether local variables are required cannot be determined until the design reaches the interplay between the physical domain and the process domain, and high-level process variables are established. For instance, whether a low-level DP will be manipulated directly or its content will be copied into a local variable is a decision to be made during zigzagging between the physical domain and the process domain.

When the selected DPs are implemented using a set of PVs (e.g., programming language, compiler), the resulting solution should satisfy the intended design goals.

This novel interpretation provides multiple advantages. First, it increases reusability by emphasizing the process domain and, consequently, the role of PVs. Without utilizing the process domain, we will end up in a recursive design loop, constantly redefining different design parameters. Second, it provides a hierarchical view of FRs, DPs, and PVs, enforcing their interpretation according to the depth decomposition. Third, this interpretation avoids enforcing a strict one-to-one correspondence between an abstract FR and a specific architectural element. In axiomatic design, a high-level FR may legitimately map to a DP interpreted as a system, a subsystem, or even a module, depending on the designer's perspective and the nature of the system under design. what matters is that the

chosen DP effectively satisfies the FR and that its level of abstraction remains consistent with the corresponding level of decomposition to avoid premature design decisions, similar to the last responsible moment principle [31].

In summary, while a flat interpretation of FRs, DPs, and PVs could confuse the designer, their rigid one-to-one mapping could likewise constrain design freedom, force premature architectural decisions, and break the alignment between abstraction level and decomposition depth.

## 7 How to Capture Input and System Constraints

How to capture input constraints (ICs) and system constraints (SCs) is another topic that deserves further study. Recall that ICs exist *before* any DP is selected and are attached directly to first-level FRs. In contrast, SCs arise *after* a DP has been chosen for an FR; their cause is the chosen DPs (e.g., technology limits, architectural consequences, physical bounds). We therefore argue that, at a given decomposition level $n$, an SC introduced by a chosen $DP_i$ should be handled at level $n + 1$ in one of two ways:

(1) as a *constraint* on the sub-FRs at level $n + 1$, or
(2) as a *new sub-FR* at level $n + 1$, that must now be satisfied.

In practice, this means that an SC at level $n$ behaves like an input constraint for the next level of decomposition. For example, consider a software price query service with the first-level FR and DP defined in Table 11. Choosing $DP_1$ introduces *staleness* as a new system constraint, resulting in:

$SC_1$: *Staleness can only be guaranteed to be $\leq 1$ s.*

At the next level of decomposition, $SC_1$ is handled by defining sub-FRs whose constraints reflect this limit.

### Table 11: Example FR and DP with sub-FRs

| FR/DP | Description |
|---|---|
| $FR_1$ | The system SHALL provide a price query service. |
| $DP_1$ | Implement a price query microservice with a distributed cache. |
| $FR_{1.1}$ | Price query service SHALL refresh cached prices [at least every 1 s]. |
| $FR_{1.2}$ | Price query service SHALL reject responses [if cache age exceeds 1 s]. |

Thus, we can conclude that $SC_1$ at the $FR_1/DP_1$ level is propagated into the next level as design ranges and constraints on $FR_{1.1}$ and $FR_{1.2}$.

## 8 Conclusion

In this study, we applied axiomatic design (AD) to end-to-end software design and offered a revised interpretation of several of its core principles. We also demonstrated the resulting artifacts in a co-engineering setting in which Copilot generated code from the specified FRs and DPs, yielding improved modularization and helping to address the limitations of LLMs4SE tools in abstract reasoning.

Another study is needed to evaluate this design in comparison with other software design patterns that have been practiced for decades and examined in empirical studies; second, it should explore more about the information axiom and investigate how it could support design rationale and reduce the complexity of code generated by LLMs.

---

[3]The C language provides basic arithmetic and logical commands, such as *MUL*, which we refer to as basic functions. However, when a designer implements a sorting function without relying on any library, that function is considered a design parameter (DP).

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
