# OpenReview forum: "Axiomatic Software Design: What GenAI Cannot Design"
_ACM.org/AIWare/2026/Conference — Submitted to AIware 2026_

### Official Review · Reviewer_9ADd · 2026-03-08

**Rating:** 3
**Confidence:** 2

**Review:**

1.	The paper identifies a gap in LLM capabilities related to abstract reasoning and their limited use in areas such as requirements engineering. To address this gap, the paper proposes incorporating Axiomatic Design into LLM-assisted software development workflows, which is an interesting topic.
2.	In Section 1, “AI-augmented software development” is described as aiming to increase developer productivity and reduce cognitive load. However, incorporating Axiomatic Design into the development process may not necessarily achieve these outcomes. For example, developers unfamiliar with AD, or LLM generating wrong code from the AD. A discussion of these potential trade-offs would strengthen the paper.
3.	In Section 1, the paper cites a 2024 survey to describe LLM usage in software engineering. Including more recent references may help provide a more up-to-date idea on current LLM adoption.
4.	In Section 1 (around line 100), the abbreviation AD is used before the term Axiomatic Design is formally introduced.
5.	There appears to be a placeholder text in the manuscript: “Section ?? outlines what axiomatic design can offer, and Section 8 concludes the paper.”
6.	The paper demonstrates the proposed approach using a calculator program. While this example helps illustrate the main idea, it would be beneficial to apply the approach on a more complex software problem which could better demonstrate its strengths.

**Summary:**

This paper explores the integration of Axiomatic Design (AD) into LLM-assisted software development. The authors present a demonstration of applying Axiomatic Design principles in the development of a calculator program to illustrate how AD may support AI-augmented software development.

---

### Official Review · Reviewer_GYdC · 2026-03-10

**Rating:** 2
**Confidence:** 3

**Review:**

The paper starts very well and it has a strong introduction framed around AI-augmented software engineering, noting that current research efforts are mainly focused on code generation (translation from natural language requirements) and that little work has been done on LLM-driven software requirements/design. Indeed, there are some studies showing that LLMs, and AI in general, do not perform very well on these abstract reasoning tasks. The motivation of the paper is strong.

Also, the use of AD is interesting. It is an old approach that had some impact in the research community but little (or no) adoption in industry. I found the overall idea appealing that AD might help LLM-assisted software engineering.

However, the paper shows no evidence that Axiomatic Design can improve LLM reasoning (besides a conjecture). This is the major issue of this work.

The paper shows an example where Copilot is used to generate C code from the final FRs and DPs. However, it does not compare this against code generated without axiomatic design or with other techniques. After reading the paper, it remains very unclear whether AD can really help LLMs reason better. The paper is only conjecturing that AD may help structure requirements and design. The rationale behind, structured decomposition, separation of <what> from <how> and traceability make some sense,  but not convincing evidence or argument, that AD translates into better LLM reasoning.

I also find the paper not very clear, especially in explaining what the insights and lessons learned are. There are the proposed FR and DP syntaxes, which appear a bit trivial as a novel contribution (and also are not evaluated). There is also an interesting discussion on creating software from needs to executable code, and the hierarchical interpretation (which makes a lot of sense). Indeed, hierarchy is key to the very well-adopted C4 model (a set of hierarchical abstractions). The paper should do a better job of making clear what is new, what is background, and what is reinterpretation. The simple example is great for grasping the fundamentals of AD, but not very good for showcasing abstract reasoning.

To conclude, the idea of re-evaluating AD and seeing if it can help AI-assisted software engineering is a very interesting idea. Unfortunately, this paper lacks evidence (even preliminary evidence would be acceptable, as AIWARE welcomes vision and new idea papers). It also lacks a clear and compelling explanation of the intuition for why AD might help LLMs.

Minor issues:
- line 99 AD acronym is used before saying what is this acronym.
- the paper should explain better that FRs and DPs don’t need to match one to one, the whats do not need to match the hows, although is desirable for traceability.

**Summary:**

The paper wants to address the issue that AI became very popular for generating code but not yet effective for more abstract parts of software engineering, like defining requirements and software design. To address this, the paper proposes using Axiomatic Design, an old design method that clearly separates what a system must do from how it will do it. Using a simple calculator example and Copilot-generated C code, the paper illustrates the process step by step. It also proposes a novel FR and DP syntaxes and an interesting hierarchical interpretation of AD.

---

> ### Author Response · Authors · 2026-03-14
> **Review feedback is interesting but it falls in side arguments of the paper**
>
> Rebuttal to Reviewer
>
> Thank you very much for your thoughtful and constructive review. I sincerely appreciate the time and effort you invested in reading the manuscript and providing detailed feedback. I am particularly grateful for your positive remarks regarding the motivation and introduction, as well as your recognition that revisiting axiomatic design (AD) in the context of AI-assisted software engineering is an interesting idea.
>
> Below I address the main concerns raised in the review.
>
> 1. Clarification of the Paper’s Core Claim
>
> One important point that may not have been sufficiently clear in the manuscript concerns the role of axiomatic design with respect to LLM reasoning.
>
> The paper does not claim that axiomatic design improves the reasoning capability of large language models themselves. Rather, the argument is that AD provides a systematic framework for structuring requirements and design artifacts—such as functional requirements (FRs), design parameters (DPs), and process variables (PVs)—which can help organize the problem space in which AI-assisted tools operate.
>
> In other words, AD does not enhance the reasoning ability of LLMs; instead, it helps structure the design process so that human designers and AI-assisted tools can interact within a clearer and more traceable design framework.
>
> 2. On the Lack of Empirical Evidence
>
> The reviewer correctly notes that the paper does not provide empirical evidence demonstrating that AD improves LLM-assisted development outcomes. This observation is valid and appreciated.
>
> The goal of this work is primarily conceptual and methodological: to explore how axiomatic design can structure software design artifacts in a way that may support AI-assisted development workflows. The calculator example is therefore intended to illustrate feasibility, demonstrating that structured FR, DP and PV artifacts derived from AD can be used to guide tools such as Copilot in generating executable code.
>
> The work should thus be viewed as an initial exploration of the idea rather than an empirical validation. I fully agree that future work should evaluate this hypothesis more rigorously, for example by comparing LLM outputs generated from:
>
> 2.1.  AD-structured design artifacts
> 2.2. traditional informal specifications
> 2.3. alternative design methods.
>
> I will clarify this positioning more explicitly in the revised manuscript.
>
> 3. Clarification of the Paper’s Contributions
>
> The reviewer also raises an important point regarding the clarity of the contributions. I tend to disagree with this point because the section 7 of the paper clearly outlines the key contributions of the paper. In addition, in the revised version I can outline the main contributions as follows:
>
> 31. End-to-end application of axiomatic design to software development, spanning the entire path from stakeholder needs to executable code.
>
> 3.2. Explicit integration of the process domain and process variables (PVs) into software design. Many previous works applying AD to software focus primarily on the functional and physical domains, whereas this work demonstrates how PVs can be incorporated into a software-centric design workflow.
>
> 3.3. A hierarchical interpretation of FRs, DPs, and PVs, clarifying how their meaning evolves across levels of abstraction during the decomposition process.
>
> 3.4. Proposed syntactic guidelines for FR and DP formulation, intended to support clearer separation between the “what” and the “how” during early stages of design.
>
> As correctly noted by the reviewer, the FR/DP syntaxes are not intended as the primary novelty, but rather as supporting mechanisms for maintaining separation between functional and physical domains.
>
> In the revised version of the paper, I will emphasize these contributions more clearly in the introduction to avoid confusion.
>
> 4. On the Simplicity of the Example
>
> The reviewer also notes that the calculator example may be too simple to demonstrate abstract reasoning capabilities.
>
> The example was intentionally chosen to keep the focus on illustrating the design methodology itself, rather than on the complexity of the system. Introducing a more complex case study could risk obscuring the step-by-step application of the axiomatic design process.
>
> That said, I agree that applying the approach to larger systems would provide stronger demonstrations of its potential, and this direction will be discussed more explicitly as future work.
>
> 5. On the Characterization of Axiomatic Design
>
> Finally, regarding the description of AD as an “old” approach, it is certainly true that axiomatic design was introduced several decades ago. However, it remains an active area of research and has been applied in several engineering domains, particularly mechanical engineering, manufacturing, and complex system design.
>
> The motivation of this paper is precisely to revisit this established design theory and explore its potential relevance in the emerging context of AI-assisted software engineering.

---

### Official Review · Reviewer_wtXq · 2026-03-11

**Rating:** 2
**Confidence:** 5

**Review:**

Pros

-  The observation that LLMs4SE research overwhelmingly focuses on code generation (\~57%)  while neglecting requirements engineering (\~4%) and software design (\~1%) is well-supported by the cited 2024 survey. Positioning AD as a complementary framework for human-AI co-engineering is a relevant contribution to the SE community.

- Authors mention that, the Prior AD-for-software work has typically stopped at high-level design. This paper goes from customer attributes to compilable C code, which is a meaningful step forward in demonstrating AD's practical applicability.

- The step-by-step decomposition is clearly presented and effectively illustrates how AD's zigzagging process works in practice, including the deliberate removal of speculative FRs , which is an instructive example of disciplined design thinking.

- The proposed hierarchical view of DPs and PVs (Section 6) is a genuine conceptual contribution that could help resolve long-standing confusion in the AD-for-software literature.

Cons

-  The calculator example is acknowledged as simple, but it is *too* simple to convincingly demonstrate that AD scales to real software systems. A calculator with four arithmetic operations does not involve concurrency, state management, external dependencies, error recovery strategies, or any of the complexities that make software design genuinely hard. The paper's central claim — that AD can help where GenAI cannot — would be far more compelling with a non-trivial example.

- The paper provides no empirical evidence that AD-guided design produces better software than alternative approaches, or that it meaningfully improves LLM-assisted design outcomes. There is no comparison with other design methodologies (e.g., Domain-Driven Design, SOLID principles, or even ad-hoc design), no user study, and no quantitative metrics (e.g., coupling, cohesion, maintainability).

- The title and abstract make a strong claim about LLM limitations in design, grounded in the ARC challenge. However, the paper does not actually test any LLM on the design task. The ARC challenge measures general abstract reasoning, not software design specifically. The gap between "LLMs struggle with ARC puzzles" and "LLMs cannot do software design" is not rigorously bridged. A simple experiment — e.g., prompting an LLM to design the same calculator and comparing the result with the AD-derived design — would have been valuable.

- The paper cites a 2024 survey on LLMs4SE but does not engage with recent work on LLM-based design assistants, prompt engineering for architecture, or AI-driven requirements elicitation. The related work is heavily focused on AD literature and does not adequately position the contribution within the broader AI-for-SE landscape.

The technical treatment of Axiomatic Design is solid,  the decomposition is done correctly and the design matrices are checked at each level. That said, the work would be much stronger with some empirical backing, even a small case study or a comparison with another design approach. It raises a natural question: how would this workflow hold up for a system with dozens of FRs, conflicting stakeholder needs, and non-functional concerns like performance or security? The calculator example sidesteps all of that. On the writing side, the paper reads well overall. The zigzagging walkthrough in Sections 4–5 is clear and the tables help a lot. The most original part is the hierarchical interpretation of DPs and PVs in Section 6, which clears up a real source of confusion in prior AD-for-software work. The FR/DP syntaxes based on ISO xx are a nice addition too, though it remains to be seen how useful they are in practice. The topic itself matters , systematic design methods are almost entirely absent from the LLMs4SE literature , and the direction is promising. A natural next step would be a controlled experiment comparing AD-guided prompts with free-form prompts when using an LLM for design, measuring things like coupling, cohesion, and correctness. Right now, though, the significance is held back by the simplicity of the example and the lack of evidence that this approach actually leads to better designs than just prompting an LLM directly. Even so, the paper does tackle a relevant problem , the mismatch between what LLMs can do and what software design actually demands , and applying Axiomatic Design end-to-end is an interesting way to address it. The hierarchical DP/PV interpretation is a genuine conceptual contribution, and the step-by-step walkthrough has real pedagogical value. The main concern is that the trivial example, the missing empirical evaluation, and the under-supported title claim make the work read more as a position paper or tutorial than a research contribution with validated results.




Minor Issues

- Table 4, FR1.3: "vlid" should be "valid"; "charachter" should be "character."
- Table 5, DP1.4: "dose" should be "does."
- Table 7, DP2.3: "exactlly" should be "exactly."
- Table 8, PV2: "evalute" should be "evaluate."
- Section 1, paragraph referencing Section 7: "Section ??" — broken cross-reference.

**Summary:**

This paper argues that current Large Language Models excel at code generation (a syntactic, translation-based task) but fundamentally struggle with software design and requirements engineering, which demand abstract thinking and abductive reasoning — capabilities where LLMs still fall short, as evidenced by the Abstract Reasoning Corpus (ARC) challenge. To address this gap, the authors propose applying Axiomatic Design (AD), a systematic design framework.

The paper makes three main contributions:

1. Demonstrates the full AD workflow, from customer needs through functional requirements, design parameters, and process variables, all the way to compilable C code, using a simple calculator example.

2. The authors propose formal syntaxes for formulating FRs and DPs,, to help designers (and LLM-based assistants) maintain a clear separation between the "what" (functional domain) and the "how" (physical domain) at each decomposition level.

3. The paper introduces a novel hierarchical view where DPs and PVs take on different meanings depending on the decomposition level (e.g., DPs are modules at the top level, algorithms at mid-level, and inputs at the leaf level).

The design is demonstrated through a calculator performing four arithmetic operations on two natural numbers. The zigzagging process between functional and physical domains is shown step by step, design matrices are evaluated against the Independence Axiom at each layer, and the resulting C code (generated by Copilot from the final FRs/DPs) is presented.

---

> ### Author Response · Authors · 2026-03-14
> **Overlooked Section and Core Concept**
>
> Rebuttal to Reviewer
>
> Thank you for the thoughtful and constructive review. I appreciate the detailed feedback and the positive assessment of the paper’s technical correctness, clarity of the decomposition process, and the identification of the hierarchical interpretation of DPs and PVs as a conceptual contribution. I address the main concerns below.
>
> 1. On the Simplicity of the Example
>
> Regarding the concern that the calculator example is too simple to demonstrate scalability to real software systems:
>
> The goal of using the calculator example is not to address challenges such as concurrency, state management, external dependencies, or other common non-functional requirements. Rather, the objective is to illustrate the axiomatic design workflow in a transparent and easily understandable way, allowing readers to focus on the methodology itself without being distracted by domain-specific complexities.
>
> Axiomatic design relies on a disciplined zigzagging process between functional and physical domains, and presenting this process clearly requires a problem that is simple enough for the reader to follow each decomposition step. For this reason, the calculator example is intentionally pedagogical.
>
> More complex applications of axiomatic design to software systems have been explored in other works by the author, and I will add references to these works in the revised manuscript to better contextualize how the methodology applies to larger systems.
>
> 2. On the Lack of Empirical Evaluation
>
> Regarding the absence of empirical evidence comparing AD-guided design with alternative approaches:
>
> The goal of this paper is not to conduct a comparative evaluation of software design methodologies, but rather to demonstrate how axiomatic design can be systematically applied to software development from customer needs to executable code (the core novelty of work).
>
> In particular, the paper focuses on showing the full AD workflow across domains (customer attributes → FRs → DPs → PVs) and illustrating how this structured design process can interface with AI-assisted code generation tools.
>
> That said, the reviewer correctly notes that complex empirical studies comparing AD-guided workflows with alternative design approaches would be valuable. However, note that the benefit of using a simple empirical example is that the reader can easily evaluate the output of the paper by using any GenAI tool or design method (SOLID) may reader wish and compare the outcomes with our results. This is a very trivial task for the readers. While, I can clarify this positioning more explicitly in the revised manuscript it is also important to remind to the reviewer that when FRs are independent then it does not matter whether you have 3 FR or dozens of FRs because you can easily scale the approach by employing IID. Therefore, the comment "It raises a natural question: how would this workflow hold up for a system with dozens of FRs, conflicting stakeholder needs" is irrelevant.
>
> 3. On the Absence of Comparisons With Other Design Methods
>
> The reviewer also suggests comparisons with other design approaches such as Domain-Driven Design, SOLID-based design practices, or ad-hoc design.
>
> Conducting such comparisons would require substantial experimental evaluation across multiple systems and design teams, which would exceed both the scope and the page limits of the current paper. Instead, the goal here is to present the methodological foundation and workflow, which can subsequently enable such comparative studies.
>
> In this sense, the work can be seen as establishing the framework necessary for future empirical investigations of AD-guided software design in both human-only and human–AI collaborative settings.
>
> 4. On the Claim About LLM Limitations
>
> Regarding the concern that the title and abstract make strong claims about LLM limitations in design while the paper does not experimentally test LLMs on design tasks:
>
> This is a fair observation. However, please note that the intent of referencing the ARC challenge and related work was to motivate the broader discussion about the limitations of current LLM systems in tasks requiring abstract reasoning and structured problem decomposition.
>
> LLMs inherently lack abductive reasoning, as shown in prior research; this paper does not aim to enhance or enable such reasoning or abstract thinking.
>
> Therefore, the primary focus of the paper is not to empirically evaluate LLM design capabilities, but rather to explore how established design methodologies such as axiomatic design may provide a structured framework for AI-assisted (i.e., AI in cooperation with human) software engineering).
>
> Finally, ensuring FRs stay independent throughout the design process is an effective way to handle conflicting stakeholder needs.
>
> To avoid potential misunderstanding, I will revise the title and abstract to better reflect the paper’s central focus on the application of axiomatic design.